# Towards Fast Adaptation of Neural Architectures with Meta Learning

**Dongze Lian**[1]*, **Yin Zheng**[2]*, **Yintao Xu**[1], **Yanxiong Lu**[2], **Leyu Lin**[2], **Peilin Zhao**[3], **Junzhou Huang**[3,4], **Shenghua Gao**[1]†

[1] ShanghaiTech University,      [2] Weixin Group, Tencent,
[3] Tencent AI Lab,      [4] University of Texas at Arlington

{liandz, xuyt, gaoshh}@shanghaitech.edu.cn, {yzheng3xg}@gmail.com,
{alanlu, goshawklin, masonzhao}@tencent.com, {jzhuang}@uta.edu

## Abstract

Recently, Neural Architecture Search (NAS) has been successfully applied to multiple artificial intelligence areas and shows better performance compared with hand-designed networks. However, the existing NAS methods only target a specific task. Most of them usually do well in searching an architecture for single task but are troublesome for multiple datasets or multiple tasks. Generally, the architecture for a new task is either searched from scratch, which is neither efficient nor flexible enough for practical application scenarios, or borrowed from the ones searched on other tasks, which might be not optimal. In order to tackle the transferability of NAS and conduct fast adaptation of neural architectures, we propose a novel Transferable Neural Architecture Search method based on meta-learning in this paper, which is termed as T-NAS. T-NAS learns a meta-architecture that is able to adapt to a new task quickly through a few gradient steps, which makes the transferred architecture suitable for the specific task. Extensive experiments show that T-NAS achieves state-of-the-art performance in few-shot learning and comparable performance in supervised learning but with 50x less searching cost, which demonstrates the effectiveness of our method.

## 1 Introduction

Deep neural networks have achieved huge successes in many machine learning tasks (Girshick, 2015; He et al., 2016; Sutskever et al., 2014; Zheng et al., 2015b; Lian et al., 2019; Cheng et al., 2019; Zheng et al., 2015a; Lauly et al., 2017; Jiang et al., 2017; Zheng et al., 2016). Behind their successes, the design of network architecture plays an important role, and the hand-designed networks (*e.g.*, ResNet (He et al., 2016), DenseNet (Huang et al., 2017)) have provided strong baselines in many tasks.

Neural Architecture Search (NAS) (Pham et al., 2018; Liu et al., 2018b; Guo et al., 2019) is proposed to automatically search network structure for alleviating the complicated network design and heavy dependence on prior knowledge. More importantly, NAS has been proved to be effective and obtained the remarkable performance in image classification (Pham et al., 2018; Liu et al., 2018b), object detection (Ghiasi et al., 2019) and semantic segmentation (Chen et al., 2018; Liu et al., 2019). However, the existing NAS methods only target a specific task. Most of them usually do well in searching an architecture for single task but are troublesome for multiple datasets or multiple tasks. As shown in Figure 1, we get the architecture-0 on a given dataset using a NAS method. Now, what if there exists a new task? This drives us to ask: how to get a suitable architecture for a new task in NAS? Generally, there exist two simple solutions in handling multiple tasks. One of them (S1) is to search an architecture for a new task from scratch but it is inefficient and not flexible for practical application scenarios. Another solution (S2) is to borrow architecture from the ones searched on other tasks but it might be not optimal for the new task. Therefore, it is urgently needed to study the transferability of NAS for large-scale model deployment in practical application. It should be

---

*Equal contribution, this work is done when Dongze Lian works as an intern in Tencent AI Lab.
†Corresponding author.

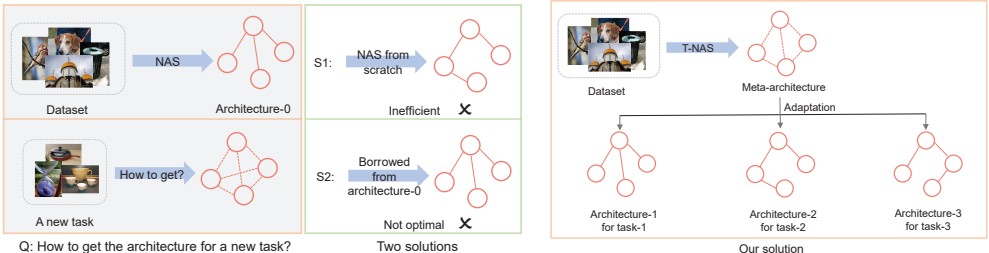

Figure 1: Left: how to search the network architecture when given a new task? Middle: two simple solutions that are inefficient or not optimal. Right: we propose T-NAS method to get a meta-architecture, which is able to adapt to different tasks easily and quickly.

more desirable to learn a transferable architecture that can adapt to some new unseen tasks easily and quickly according to the previous knowledge.

To this end, we propose a novel Transferable Neural Architecture Search (T-NAS) method (the bottom of Figure 1). The starting point of T-NAS is inspired by recent meta-learning methods (Finn et al., 2017; Antoniou et al., 2019; Sun et al., 2019), especially Model-Agnostic Meta-Learning (MAML) (Finn et al., 2017), where a model learns the meta-weights that are able to adapt to a new task through a few gradient steps. Push it forward, it is also possible to find a good initial point of network architecture for NAS. Therefore, the T-NAS learns a meta-architecture (transferable architecture) that is able to adapt to a new task quickly through a few gradient steps, which is more flexible than other NAS methods. Similar to MAML, such a good initial meta-architecture for adaptation should be more sensitive to changes in different tasks such that it can be easily transferred. It is worth mentioning that this is not the first work on the transferability of neural architecture. There are also some recent works that attempt to utilize the knowledge on neural architectures learned from previous tasks, such as Wong et al. (2018); Shaw et al. (2018). Specifically, Wong et al. (2018) proposes to transfer the architecture knowledge under a multi-task learning perspective, where the number of tasks is fixed during training phase, and it cannot do a fast adaption for a new task. In contrast, our model is able to make the adaption fast and the number of tasks is unlimited during training. The difference between our model and Shaw et al. (2018) is also obvious, where Shaw et al. (2018) is based on Bayesian inference but our model is based on gradient-based meta-learning. The quantitative comparison with Shaw et al. (2018) can be found in Table 3.

Generally, architecture structure cannot be trained independently regardless of network weights (Liu et al., 2018b; Pham et al., 2018). Analogously, the training of meta-architecture is also associated with meta-weights. Therefore, the meta-architecture and meta-weights need to be optimized jointly across different tasks, which is a typical bilevel optimization problem (Liu et al., 2018b). In order to solve the costly bilevel optimization in T-NAS, we propose an efficient first-order approximation algorithm to update meta-architecture and meta-weights together. After the whole model is optimized, given a new task, we can get the network architecture structure suitable for the specific task with a few gradient steps from meta-architecture and meta-weights. At last, the decoded discrete architecture is used for the final architecture evaluation.

To demonstrate the effectiveness of T-NAS, we conduct extensive experiments on task-level problems due to amounts of tasks. Specifically, we split the experiments into two parts: few-shot learning setting and supervised learning setting. For few-shot learning, T-NAS achieves state-of-the-art performance on multiple datasets (Omniglot, Mini-Imagenet, Fewshot-CIFAR100) compared with previous methods and other NAS-based methods. As for supervised learning, a 200-shot 50-query 10-way experiment setting is designed on the Mini-Imagenet dataset. Compared with the searched architectures from scratch for new given tasks, T-NAS achieves comparable performance but with 50x less searching cost.

Our main contributions are summarized as follows:

- We propose a novel Transferable Neural Architecture Search (T-NAS). T-NAS can learn a meta-architecture that is able to adapt to a new task quickly through a few gradient steps, which is more flexible than other NAS methods.

- We give the formulation of T-NAS and analyze the difference between T-NAS and other NAS methods. Further, to solve the bilevel optimization, we propose an efficient first-order approximation algorithm to optimize the whole search network based on gradient descent.

- Extensive experiments show that T-NAS achieves state-of-the-art performance in few-shot learning and comparable performance in supervised learning but with 50x less searching cost, which demonstrates the effectiveness of our method.

## 2 RELATED WORK

### 2.1 NEURAL ARCHITECTURE SEARCH

Neural Architecture Search (NAS) designs network architectures automatically instead of hand-designed ones. Generally, NAS strategies are divided into three categories - reinforcement learning, evolutionary algorithm and gradient-based methods. Some other strategies can refer to the survey paper (Elsken et al., 2019). Reinforcement learning (RL) based methods (Zoph & Le, 2016; Zoph et al., 2018) utilize a controller to generate the network structure and operations. For efficient searching, ENAS (Pham et al., 2018) shares parameters among child models and achieves state-of-the-art performance with only one GPU day. Evolutionary algorithm based methods (Real et al., 2018) evolve neural architectures and also achieve comparable results with RL based methods.

Unlike reinforcement learning and evolutionary algorithm, gradient-based methods (Liu et al., 2018b; Cai et al., 2019) continuously relax the discrete architecture with all possible operations, which makes it possible to jointly optimize the architecture structure and network weights based on gradient descent. Not limited to image classification problems, recent works also introduce NAS to object detection (Ghiasi et al., 2019) and semantic image segmentation (Chen et al., 2018; Liu et al., 2019). More recently, NAS is also applied to the generative model, such as AutoGAN (Gong et al., 2019). These NAS methods show that the searched networks outperform the hand-designed ones.

However, in these methods, only a fixed architecture is searched for a specific task, which makes it hard to be transferred to other tasks. In order to obtain a more flexible network, InstaNAS (Cheng et al., 2018) is proposed to search the network architecture structure for each instance according to different objectives, such as accuracy or latency. Different from Cheng et al. (2018), we incorporate the ideas from meta-learning based methods and extend NAS to T-NAS, which learns a meta-architecture that is able to adapt to different tasks.

### 2.2 FEW-SHOT META-LEARNING

Recently, most of few-shot learning problems can be cast into the meta-learning field, where a model is trained to quickly adapt to a new task given only a few samples (Finn et al., 2017). Such few-shot meta-learning methods can be categorized into metric learning (Vinyals et al., 2016; Sung et al., 2018; Snell et al., 2017), memory network (Santoro et al., 2016; Oreshkin et al., 2018; Munkhdalai et al., 2018; Mishra et al., 2018) and gradient-based methods (Finn et al., 2017; Zhang et al., 2018; Sun et al., 2019).

Here, we only focus on the gradient-based methods, which contain a base-learner and a meta-learner. MAML (Finn et al., 2017) is one of the typical gradient-based methods for fast adaptation, which consists of meta-train and meta-test stages. In the meta-train stage, the model extracts general knowledge (meta-weights) from amounts of tasks such that it can be utilized for fast adaptation in the meta-test stage. The latest variant of MAML is MAML++ (Antoniou et al., 2019), which analyzes the shortcoming of MAML and proposes some tips on how to train MAML to promote the performance. We extend the adaptation of weights in MAML to the adaptation of architectures that is also based on MAML, and propose to automatically learn a meta-architecture, which is able to adapt to different tasks quickly.

## 3 PRELIMINARY

To introduce T-NAS, we briefly review the knowledge about meta-learning for fast adaptation (Finn et al., 2017; Antoniou et al., 2019) and DARTS for NAS (Liu et al., 2018b) in this section, which is helpful to understand the concept of T-NAS.

### 3.1 META-LEARNING

The whole dataset, meta-train and meta-test dataset are denoted as $\mathcal{D}$, $\mathcal{D}_{\text{meta-train}}$ and $\mathcal{D}_{\text{meta-test}}$, respectively. In meta-train stage, a set of tasks $\{\mathcal{T}\}$ (are also called episodes) are sampled from the task distribution $p(\mathcal{T})$ in $\mathcal{D}_{\text{meta-train}}$. Note that in the $i$-th task $\mathcal{T}_i$, there are $K$ samples from each class and $N$ classes in total, which is typically formulated as a $N$-way, $K$-shot problem. The training split samples in $\mathcal{T}_i$ used to optimize the base-learner are called support set, denoted as $\mathcal{T}_i^s$, and test split samples used to optimize the meta-learner are called query set, which is $\mathcal{T}_i^q$. The main idea of MAML (Finn et al., 2017) is to learn good initialized weights $\widetilde{w}$ for all tasks $\{\mathcal{T}\}$, such that the network can obtain high performance in $\mathcal{D}_{\text{meta-test}}$ after a few gradient descent steps from $\widetilde{w}$. The base-learner is optimized according to the following rule:

$$w_i^{m+1} = w_i^m - \alpha_{\text{inner}} \nabla_{w_i^m} \mathcal{L}(f(\mathcal{T}_i^s; w_i^m)), \tag{1}$$

where $\alpha_{\text{inner}}$ is the inner (base) learning rate of weights $w$ and $m$ represents the inner step. $f$ is the parametrized function with network weights $w$ and $\mathcal{L}$ is the loss function. In the base-learner process, $\mathcal{T}_i^s$ is used to compute the loss and we update weights $w$ from $w_i^m$ to $w_i^{m+1}$ for the $i$-th task ($w_i^0 = \widetilde{w}$). After $M$ steps, $\mathcal{L}(f(\mathcal{T}_i^q; w_i^M))$ in $\mathcal{T}_q$ is computed for the meta-learner update, which can be formulated as:

$$\widetilde{w} = \widetilde{w} - \alpha_{\text{outer}} \nabla_{\widetilde{w}} \sum_{\mathcal{T}_i^q \sim p(\mathcal{T})} \mathcal{L}(f(\mathcal{T}_i^q; w_i^M)), \tag{2}$$

where $\alpha_{\text{outer}}$ is the outer (meta) learning rate of meta-weights $\widetilde{w}$. Finally, the model learns the good initialized meta-weights $\widetilde{w}$ when it converges. Such meta-weights are sensitive enough so that it can adapt to each task in $\mathcal{D}_{\text{meta-test}}$ after a few gradient descent steps.

### 3.2 DARTS

The core of DARTS (Liu et al., 2018b) is to continuously relax the discrete architecture with all possible operations and jointly optimize the architecture structure and network weights based on gradient descent. Consider $\mathcal{O}$ be a set of candidate operations, where each operation proposal is represented with $o$. Given the input $x$, the output is the weighted sum of all possible operations $o(x)$:

$$\overline{o}(x) = \sum_{o \in \mathcal{O}} \frac{exp(\theta_o)}{\sum_{o' \in \mathcal{O}} exp(\theta_{o'})} o(x), \tag{3}$$

where $\theta$ is the vector to represent the coefficients of different operation branches. When decoding, the operation $o^* = \arg\max_{o \in \mathcal{O}} \theta_o$. Therefore, $\theta$ is also the encoding of the architecture.

To solve such a bi-level optimization problem, a two-step update algorithm is applied:

$$\begin{cases} \theta = \theta - \beta \nabla_\theta \mathcal{L}(w - \xi \nabla_w \mathcal{L}(w, \theta), \theta) \\ w = w - \alpha \nabla_w \mathcal{L}(w, \theta) \end{cases}, \tag{4}$$

where $\mathcal{L}$ is the loss function and $\xi$ is the learning rate of inner optimization. In this paper, we use the first-order optimization of DARTS ($\xi = 0$) for efficiency.

## 4 APPROACH

In this section, we first introduce Transferable Neural Architecture Search (T-NAS) and give its formulation. After that, we analyze and illustrate the difference between T-NAS and NAS. Finally, the first-order approximation algorithm is proposed for the optimization of T-NAS, and the adaptation and decoding process are also described in detail.

### 4.1 THE FORMULATION OF T-NAS

To make our searched network architecture flexible, we focus on the transferability of NAS. As shown in Sec. 3, MAML is trained to learn meta-weights $\widetilde{w}$ for fast adaptation in a new task. Similarly, T-NAS devotes itself to learn a meta-architecture $\widetilde{\theta}$ that is able to adapt to a new task through a few steps. In this work, $\theta$ and $\widetilde{\theta}$ [1] are defined as the encoding of the architecture and transferable architecture, which are represented as matrices following DARTS (Liu et al., 2018b).

To make the searched architecture transferable, we utilize the meta-learning based strategy to learn a task-sensitive meta-architecture $\widetilde{\theta}$. However, similar to other NAS methods (Pham et al., 2018; Liu et al., 2018b), where the architecture $\theta$ usually cannot be trained independently regardless of network weights $w$, the training of meta-architecture $\widetilde{\theta}$ is also associated with meta-weights $\widetilde{w}$. In this work, $\widetilde{\theta}$ and $\widetilde{w}$ are optimized jointly across different tasks in T-NAS.

As shown in Sec. 3, there exist two learners for the learning of meta-weights $\widetilde{w}$, *i.e.*, Eq. (1) is used to update the base-learner and Eq. (2) is used to update the meta-learner. Similarly, T-NAS consists of two searchers: base-searcher and meta-searcher. In the base-searcher, $\theta$ and $w$ are optimized jointly to search architecture for the specific task $\mathcal{T}_i^s$, which can be optimized with:

$$
\begin{cases}
w_i^{m+1} = w_i^m - \alpha_{\text{inner}}\nabla_{w_i^m}\mathcal{L}(g(\mathcal{T}_i^s; \theta_i^m, w_i^m)) \\
\theta_i^{m+1} = \theta_i^m - \beta_{\text{inner}}\nabla_{\theta_i^m}\mathcal{L}(g(\mathcal{T}_i^s; \theta_i^m, w_i^{m+1}))
\end{cases}
, \qquad (5)
$$

where $\beta_{\text{inner}}$ is the inner (base) learning rate of architecture $\theta$. $g$ is the parametrized function with the architecture $\theta$ and network weights $w$ ($\theta_i^0 = \widetilde{\theta}, w_i^0 = \widetilde{w}$). After $M$ steps, $\widetilde{\theta}$ and $\widetilde{w}$ are also updated to get a good initial point for architecture adaptation in the meta-searcher, where $\mathcal{L}(g(\mathcal{T}_i^q; \theta_i^M, w_i^M))$ in $\mathcal{T}_i^q$ is computed. The formulation can be represented as:

$$
\begin{cases}
\widetilde{w} = \widetilde{w} - \alpha_{\text{outer}}\nabla_{\widetilde{w}} \sum_{\mathcal{T}_i^q \sim p(\mathcal{T})} \mathcal{L}(g(\mathcal{T}_i^q; \theta_i^M, w_i^M)) \\
\widetilde{\theta} = \widetilde{\theta} - \beta_{\text{outer}}\nabla_{\widetilde{\theta}} \sum_{\mathcal{T}_i^q \sim p(\mathcal{T})} \mathcal{L}(g(\mathcal{T}_i^q; \theta_i^M, w_i^M))
\end{cases}
, \qquad (6)
$$

where $\beta_{\text{outer}}$ is the outer (meta) learning rate of the meta-architecture $\widetilde{\theta}$. When the meta-searcher converges, the optimal meta-architecture $\widetilde{\theta}$ and meta-weights $\widetilde{w}$ can be obtained. We argue that such a $\widetilde{\theta}$ can quickly adapt to a new task. The complete algorithm of T-NAS is as shown in Alg. 1.

---

**Algorithm 1:** T-NAS: Transferable Neural Architecture Search

---

**Input:** Meta-train dataset $\mathcal{D}_{\text{meta-train}}$, learning rate $\alpha_{\text{inner}}, \alpha_{\text{outer}}, \beta_{\text{inner}}$ and $\beta_{\text{outer}}$.

1   Randomly initialize architecture parameter $\theta$ and network weights $w$.
2   **while** *not done* **do**
3     Sample batch of tasks $\{\mathcal{T}\}$ in $\mathcal{D}_{\text{meta-train}}$;
4     **for** $\mathcal{T}_i \in \{\mathcal{T}\}$ **do**
5       Get datapoints $\mathcal{T}_i^s$;
6       Compute $\mathcal{L}(g(\mathcal{T}_i^s; \theta_i^m, w_i^m))$ according to the standard cross-entropy loss;
7       Alternatively update $w_i^m$ and $\theta_i^m$ with Eq. (5) for $M$ steps;
8       Get datapoints $\mathcal{T}_i^q$ for meta-searcher;
9     **end**
10    Alternatively update $\widetilde{w}$ and $\widetilde{\theta}$ with Eq. (6);
11   **end**

---

[1] It is worth noting that the transferability of architecture is a generalized concept, which is not limited to the representation of architecture. T-NAS employs DARTS (Liu et al., 2018b) for NAS but other representations of architectures such as ENAS (Pham et al., 2018) can also be adopted.

Table 1: The main differences among NAS, Solution1 (S1), Solution2 (S2) and T-NAS.

| Methods | Task(s) | Transferability | Characteristic |
|---------|---------|-----------------|----------------|
| NAS | single | no | troublesome for multiple tasks |
| S1 | multiple | no (search from scratch) | inefficient & time-consuming |
| S2 | multiple | borrows from searched architecture | not optimal |
| T-NAS | multiple | adaptation | flexible |

## 4.2 T-NAS VS. NAS

As mentioned before, the previous NAS methods usually do well in searching an architecture for a single task but are troublesome for multiple datasets or multiple tasks. So we focus on the transferability of NAS across multiple tasks in this paper. Two simple solutions (S1 and S2) have been pointed in Figure 1 but they are either inefficient or not optimal. T-NAS aims to learn a transferable and flexible architecture that can adapt to a new task easily. Table 1 lists the main differences among NAS, two simple solutions (S1 and S2) and T-NAS. S1 does not study the transferability of NAS and searches architectures for different tasks (*e.g.*, $\theta_1, \theta_2, ..., \theta_n$) from scratch. S2 borrows from searched architecture directly such that all tasks share the same architecture (*e.g.*, $\theta$). Differently, T-NAS searches the meta-architecture $\widetilde{\theta}$, which is able to adapt to different tasks quickly (*e.g.*, $\widetilde{\theta} \to \theta_1, \theta_2, ..., \theta_n$). The experimental results show that our method achieves better performance than the S2 and comparable performance with S1 but with the less searching cost.

It is worth mentioning that if directly apply NAS to few-shot meta-learning, *e.g.*, MAML (Finn et al., 2017), we will search a good network architecture for MAML, which is named Auto-MAML. In fact, Auto-MAML is a special case of S2 in Figure 1, where all tasks share the same architecture searched with a meta-learning method. In the experiments in few-shot learning, we also introduce Auto-MAML as a baseline. However, such a shared architecture is not suitable for each task. Auto-MAML can outperform MAML but is inferior to T-NAS. The specific algorithm and experimental settings of Auto-MAML are provided in the supplementary material.

The core of T-NAS is based on MAML (Finn et al., 2017), which is a kind of gradient-based meta-learning method. Recently, MAML++ is proposed by Antoniou et al. (2019), which introduces several techniques [2] to improve the performance of MAML. These techniques can also be utilized by T-NAS, which is termed as T-NAS++ in this paper. The experiments in Section 5 confirm that T-NAS++ can further improve the performance of T-NAS.

## 4.3 OPTIMIZATION

Although the formulation of T-NAS is proposed, the model is hard to be optimized directly according to Alg. 1. On one hand, updating $\widetilde{\theta}$ and $\widetilde{w}$ introduces the high-order derivative in Eq. (6). On the other hand, the continuous relaxation of architecture makes amounts of memory occupied. At the first glance, such a problem might be solved by the first-order approximation in Liu et al. (2018b), however, there still exists a lot of time overhead, even the experiments cannot be carried out when step $M$ is large in Eq. (6). To tackle this problem, we transform the alternative update strategy of $w$ and $\theta$ in Eq. (5) into simultaneous update, which means the $w$ and $\theta$ are treated equally as the parameters of function $g$. Such a replacement can update parameters ($w$ and $\theta$) by only backpropagating once instead of twice. The Eq. (5) can be modified to:

$$[w_i^{m+1}; \theta_i^{m+1}] = [w_i^m; \theta_i^m] - \boldsymbol{\eta}_{\text{inner}} \nabla_{[w_i^m, \theta_i^m]} \mathcal{L}(g(\mathcal{T}_i^s; \theta_i^m, w_i^m)), \tag{7}$$

where $\boldsymbol{\eta}_{\text{inner}} = [\alpha_{\text{inner}}; \beta_{\text{inner}}]$. In addition, to avoid the high-order derivative, we also utilize the first-order approximation to compute the derivation of $w_i^M$ and $\theta_i^M$ instead of $\widetilde{w}$ and $\widetilde{\theta}$ as follows:

$$[\widetilde{w}; \widetilde{\theta}] = [\widetilde{w}; \widetilde{\theta}] - \boldsymbol{\eta}_{\text{outer}} \sum_{\mathcal{T}_i \sim p(\mathcal{T})} \nabla_{[w_i^M, \theta_i^M]} \mathcal{L}(g(\mathcal{T}_i^q; \theta_i^M, w_i^M)), \tag{8}$$

---

[2]These techniques include cosine annealing of the meta-optimizer learning rate, the adding of inner steps *etc.*.

where $\boldsymbol{\eta}_{\text{outer}} = [\alpha_{\text{outer}}; \beta_{\text{outer}}]$. Such modifications save more than half of the search time and memory while maintaining comparable performance. Thus, we can use the Eq. (7) and Eq. (8) to replace the Eq. (5) and Eq. (6) in line 7 and line 10 of Alg. 1 to update $\theta$ and $w$ in the implementation.

## 4.4 ADAPTATION AND DECODING

Once $\widetilde{\theta}$ and $\widetilde{w}$ are obtained by training the base-searcher and the meta-searcher with the first-order approximation of Alg. 1, we can adapt them to the $i$-th task and get the task-specific architecture $\theta_i^*$ for the specific task $\mathcal{T}_i$ according to the following Alg. 2.

---

**Algorithm 2:** Adaptation and decoding

**Input:** Meta-test dataset $\mathcal{D}_{\text{meta-test}}$, learning rate $\alpha_{\text{inner}}$ and $\beta_{\text{inner}}$.
**Output:** The task-specific architecture $\theta_i^*$ for the $i$-th task $\mathcal{T}_i$.
1 Obtain the specific task $\mathcal{T}_i$ from $\mathcal{D}_{\text{meta-test}}$;
2 Update $w_i^m$ and $\theta_i^m$ for $M$ step with Eq. (7) and get $\theta_i^M$;
3 Decoding $\theta_i^M$ to task-specific architecture $\theta_i^*$ by following the method in Liu et al. (2018b).

---

Following previous NAS methods (Zoph & Le, 2016; Zoph et al., 2018; Pham et al., 2018; Liu et al., 2018b), after getting $\theta_i^*$, we evaluate the task-specific architecture by training it in the task $\mathcal{T}_i$ from scratch. As shown in Sec. 5, the T-NAS achieves state-of-the-art performance in few-shot learning and comparable performance in supervised learning but with less searching cost.

## 5 EXPERIMENTS

We evaluate the effectiveness of T-NAS in both few-shot and supervised learning settings, as well as multiple datasets. For each dataset, we conduct experiments containing architecture search and architecture evaluation. In the architecture search stage, we use T-NAS to search for a meta-architecture. In the architecture evaluation stage, we evaluate the transferred task-specific architectures by training them from scratch and compare their performance with previous methods. S1 and S2 in the following sections mean two simple solutions in Figure 1 except for the specific instructions. Code is available [3].

### 5.1 DATASETS

**Omniglot** is a handwritten character recognition dataset proposed in Lake et al. (2011), which contains 1623 characters with 20 samples for each class. We randomly split 1200 characters for training and the remaining for testing, and augment the Omniglot dataset by randomly rotating multiples of 90 degrees following (Santoro et al., 2016).

**Mini-Imagenet** dataset is sampled from the original ImageNet (Deng et al., 2009). There are 100 classes in total with 600 images for each class. All images are down-sampled to $84 \times 84$ pixels and the whole dataset consists of 64 training classes, 16 validation classes and 20 test classes.

**Fewshot-CIFAR100 (FC100)** dataset is proposed in Oreshkin et al. (2018), which is based on a popular image classification dataset CIFAR100. It is more challenging than the Mini-Imagenet due to the low resolution. Following Oreshkin et al. (2018), FC100 is divided into 60 classes belonging to 12 superclasses for training, 20 classes belonging to 4 superclasses for validation and testing.

### 5.2 T-NAS FOR FEW-SHOT LEARNING

#### 5.2.1 ARCHITECTURE SEARCH.

We first get the meta-architecture $\widetilde{\theta}$ by optimizing the search network with first-order approximation of Alg. 1. In the architecture search stage, we employ the same operations as Liu et al. (2018b): $3 \times 3$ and $5 \times 5$ separable convolutions, $3 \times 3$ and $5 \times 5$ dilated separable convolutions, $3 \times 3$ max

---

[3]https://github.com/dongzelian/T-NAS

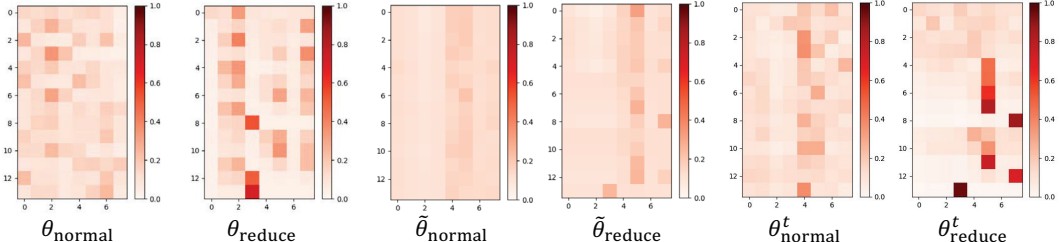

Figure 2: Architecture $(\theta_{\text{normal}}, \theta_{\text{reduce}})$ searched with Auto-MAML (left), meta-architecture $(\widetilde{\theta}_{\text{normal}}, \widetilde{\theta}_{\text{reduce}})$ searched with T-NAS (middle), and the transferred architecture $(\theta^t_{\text{normal}}, \theta^t_{\text{reduce}})$ for the specific task $\mathcal{T}_t$ (right). The experiments are conducted in 5-way, 5-shot setting of Mini-Imagenet.

pooling, $3 \times 3$ average pooling, identity and zero. ReLU-Conv-BN order is used for convolutional operations and each separable convolution is applied twice following (Liu et al., 2018a;b). For all datasets, we only use *one* {normal + reduction} cell for efficiency and preventing overfitting, thus the meta-architecture $\widetilde{\theta}$ is determined by $(\widetilde{\theta}_{\text{normal}}, \widetilde{\theta}_{\text{reduce}})$. Once $\widetilde{\theta}$ is obtained using T-NAS, we can obtain the optimal architecture $\theta^*_i$ for the specific task $\mathcal{T}_i$ from Alg. 2.

We utilize the training and validation data of dataset for architecture search. In N-way, K-shot setting, we firstly randomly sample N classes from the training classes, and then randomly sample K images for each class to get a task. Thus, there are $N \times K$ images in each task. On the Mini-imagenet dataset, *One* {normal + reduction} cell is trained for 10 epochs with 5000 independent tasks for each epoch and the initial channel is set as 16. For the base-searcher, we use the vanilla SGD to optimize the network weights $w^m_i$ and architecture parameter $\theta^m_i$ with inner learning rate $\alpha_{\text{inner}} = 0.1$ and $\beta_{\text{inner}} = 30$. The inner step $M$ is set as 5 for the trade-off between accuracy and efficiency. For the meta-searcher, we use the Adam (Kingma & Ba, 2014) to optimize the meta-architecture $\widetilde{\theta}$ and network weights $\widetilde{w}$ with outer learning rate $\alpha_{\text{outer}} = 10^{-3}$ and $\beta_{\text{outer}} = 10^{-3}$. All search and evaluation experiments are performed using NVIDIA P40 GPUs. The whole search process takes about 2 GPU days.

In addition, we also conduct Auto-MAML experiments where all tasks share the same searched architecture. Auto-MAML is a special case of S2 of Figure 1, where all tasks share the same architecture searched with a meta-learning method. In the practical algorithm, it is similar to T-NAS, which is behaved as removing the update for $\theta$ in the meta-searcher stage. However, in Auto-MAML, we can divide the whole dataset into two splits for the updates of $\theta$ and $\widetilde{w}$ following the recent gradient-based NAS methods (Pham et al., 2018; Liu et al., 2018b). Here, the $\mathcal{D}_{\text{meta-train}}$ is divided into two independent splits $\mathcal{D}_{\text{train-split1}}$ and $\mathcal{D}_{\text{train-split2}}$ with 1 : 1. The specific algorithm for meta-train and meta-test and searched architecture structure can be found in the supplementary material.

To show the transferability of meta-architecture, we visualize the (encoding of) architecture $\theta$ searched with Auto-MAML, meta-architecture $\widetilde{\theta}$ searched with T-NAS, and transferred architecture $\theta^t$ for a specific task $\mathcal{T}_t$ in Figure 2. It is worth noting that the architecture encoding matrix $(\widetilde{\theta}_{\text{normal}}, \widetilde{\theta}_{\text{reduce}})$ searched with T-NAS is smoother than that with Auto-MAML, which implies that $(\widetilde{\theta}_{\text{normal}}, \widetilde{\theta}_{\text{reduce}})$ is easier to adapt to the specific task $(\widetilde{\theta} \to \theta^t)$ than Auto-MAML, thus the meta-architecture searched with T-NAS is more flexible.

### 5.2.2 ARCHITECTURE EVALUATION.

After getting the architecture structure $\theta^*_i$ for task $\mathcal{T}_i$, we evaluate $\theta^*_i$ by training it from scratch. In architecture evaluation, we train the task-specific architecture for 20 epochs with 15000 independent tasks for each epoch. Note that different from Liu et al. (2018b), we directly use the searched network structure to evaluate performance without any modification (*e.g.*, the number of channels or layers). We optimize the network weights $w^m_i$ with $\alpha_{\text{inner}} = 0.1$ and $M = 5$. We use Adam (Kingma & Ba, 2014) to optimize the meta-weights $\widetilde{w}$ with outer learning rate $\alpha_{\text{outer}} = 10^{-3}$. The experimental results on Omniglot, Mini-Imagenet and FC100 are shown in Table. 2, Table. 3 and Table. 4, respectively, where T-NAS is based on first-order MAML. Specifically, T-NAS outperforms

Table 2: 5-way accuracy results on the Omniglot dataset.

| Methods | 1-shot | 5-shot |
|---|---|---|
| Siamese Nets (Koch et al., 2015) | 97.3% | 98.4% |
| Matching nets (Vinyals et al., 2016) | 98.1% | 98.9% |
| Neural statistician (Edwards & Storkey, 2017) | 98.1% | 99.5% |
| Memory Mod. (Kaiser et al., 2017) | 98.4% | 99.6% |
| Meta-SGD (Li et al., 2017) | **99.53 ± 0.26%** | 99.93 ± 0.09% |
| MAML (Finn et al., 2017) | 98.7 ± 0.4% | 99.9 ± 0.1% |
| MAML++ (Antoniou et al., 2019) | 99.47% | 99.93% |
| Auto-MAML (ours) | 98.95 ± 0.38% | 99.91 ± 0.09% |
| T-NAS (ours) | 99.16 ± 0.34% | 99.93 ± 0.07% |
| **T-NAS++ (ours)** | 99.35 ± 0.32% | **99.93 ± 0.07%** |

Table 3: 5-way accuracy results on Mini-Imagenet.

| Methods | Arch. | #Param. | 1-shot | 5-shot |
|---|---|---|---|---|
| Matching nets (Vinyals et al., 2016) | 4CONV | 32.9K | 43.44 ± 0.77% | 55.31 ± 0.73% |
| ProtoNets (Snell et al., 2017) | 4CONV | 32.9K | 49.42 ± 0.78% | 68.20 ± 0.66% |
| Meta-LSTM (Ravi & Larochelle, 2017) | 4CONV | 32.9K | 43.56 ± 0.84% | 60.60 ± 0.71% |
| Bilevel (Franceschi et al., 2018) | 4CONV | 32.9K | 50.54 ± 0.85% | 64.53 ± 0.68% |
| CompareNets (Sung et al., 2018) | 4CONV | 32.9K | 50.44 ± 0.82% | 65.32 ± 0.70% |
| LLAMA (Grant et al., 2018) | 4CONV | 32.9K | 49.40 ± 1.83% | - |
| MAML (Finn et al., 2017) | 4CONV | 32.9K | 48.70 ± 1.84% | 63.11 ± 0.92% |
| MAML (first-order) (Finn et al., 2017) | 4CONV | 32.9K | 48.07 ± 1.75% | 63.15 ± 0.91% |
| MAML++ (Antoniou et al., 2019) | 4CONV | 32.9K | 52.15 ± 0.26% | 68.32 ± 0.44% |
| Auto-Meta (small) (Kim et al., 2018) | Cell | 28/28 K | 49.58 ± 0.20% | 65.09 ± 0.24% |
| Auto-Meta (large) (Kim et al., 2018) | Cell | 98.7/94.0 K | 51.16 ± 0.17% | 69.18 ± 0.14% |
| BASE (Softmax) (Shaw et al., 2018) | Cell | 1200K | - | 65.40 ± 0.74% |
| BASE (Gumbel-Softmax) (Shaw et al., 2018) | Cell | 1200K | - | 66.20 ± 0.70% |
| Auto-MAML (ours) | Cell | 23.2/26.1 K | 51.23 ± 1.76% | 64.10 ± 1.12% |
| T-NAS (ours) | Cell | 24.3/26.5 K$^\star$ | 52.84 ± 1.41% | 67.88 ± 0.92% |
| **T-NAS++ (ours)** | Cell | 24.3/26.5 K$^\star$ | **54.11 ± 1.35%** | **69.59 ± 0.85%** |

$^\star$ means the average parameters of architectures for evaluation.

MAML and Auto-MAML (52.84% vs. 48.70%, 51.23%), which validates the advantage of T-NAS. It also achieves better performance than other architecture transfer methods (*e.g.*, BASE (Shaw et al., 2018)). Actually, since the advantage of T-NAS is that the meta-architecture could adapt to a new task rather than using a fixed architecture like MAML and Auto-MAML, it usually has an additional time cost for the adaption. Usually, the adaptation procedure costs about 1.5 seconds (1-shot) and 7.8 seconds (5-shot), which is negligible compared with the improvement of accuracy. Moreover, we can also see that T-NAS++, which is an improved version of T-NAS described in Sec.4.2, achieves the best performance among all the baselines.

## 5.3 T-NAS FOR SUPERVISED LEARNING

Besides few-shot learning classification, we also conduct experiments on Mini-Imagenet for general supervised learning. Different from few-shot learning, the architecture can be searched and trained for each task due to the sufficient samples, which can be regarded as S1 in Figure 1. Due to the lack of baselines in the supervised learning setting, we choose 10 tasks with 200-shot 50-query 10-way for each task based on the Mini-Imagenet dataset for meaningful experiments.

In the experiments of supervised learning, we follow the same setting as few-shot learning for transferable architecture search. The difference is that we can train each task independently from scratch in architecture evaluation. For 10 tasks in supervised learning, we train the task-specific architecture for 200 epochs with cosine schedule, where the initial learning rate is 0.05. We use the SGD with momentum 0.9 to optimize the network weights and crop the original image and flip randomly for data argumentation.

Table 4: 5-way accuracy results on FC100.

| Methods | 1-shot | 5-shot | 10-shot |
|---|---|---|---|
| MAML (Finn et al., 2017) | $38.1 \pm 1.7\%$ | $50.4 \pm 1.0\%$ | $56.2 \pm 0.8\%$ |
| MAML++ (Antoniou et al., 2019) | $38.7 \pm 0.4\%$ | $52.9 \pm 0.4\%$ | $58.8 \pm 0.4\%$ |
| Auto-MAML (ours) | $38.8 \pm 1.8\%$ | $52.2 \pm 1.2\%$ | $57.5 \pm 0.8\%$ |
| T-NAS (ours) | $39.7 \pm 1.4\%$ | $53.1 \pm 1.0\%$ | $58.9 \pm 0.7\%$ |
| **T-NAS++ (ours)** | **$40.4 \pm 1.2\%$** | **$54.6 \pm 0.9\%$** | **$60.2 \pm 0.7\%$** |

Table 5: 200-shot, 50-query, 10-way accuracy results of supervised learning on Mini-Imagenet.

| Methods | 200-shot | Time |
|---|---|---|
| Random | $61.20 \pm 0.09\%$ | N/A |
| S1 | $64.84 \pm 0.04\%$ | 266 min |
| S2 | $62.99 \pm 0.05\%$ | N/A |
| **T-NAS (ours)** | **$64.23 \pm 0.05\%$** | **5 min** |

The experimental results in the supervised learning setting are shown in Table. 5. In S1, we search the architecture for each of 10 tasks from scratch and evaluate them. For S2, we directly use five architectures searched respectively in five different tasks (sampled with 200-shot 50-query 10-way for each task in the meta-train dataset) for the evaluation in 10 tasks. For a fair comparison, we also pick five architectures randomly from search space for each task, evaluate them in the specific task, and report their average results. It is worth noting that it does not consume searching time to randomly generate architectures or directly use the prepared architectures searched in other tasks. Thus, the time of Random and Method2 in Table. 5 is not applicable. Our T-NAS can learn a meta-architecture $\widetilde{\theta}$ and get the task-specific architecture by only updating several steps from $\widetilde{\theta}$ instead of the shared architecture. Thus, T-NAS obtains better performance than random architectures and S2 ($64.23\%$ vs. $61.20\%$, $62.99\%$). In addition, T-NAS achieves competitive performance with S1 but with 50x less time cost (5 min vs. 266 min). The fact that the performance of S1 is superior to that of T-NAS slightly is because S1 directly searches network architecture for different tasks from scratch, which is laborious as well as time-consuming. On the contrary, T-NAS can adapt to different tasks quickly by finding a good initial point $\widetilde{\theta}$, which avoids laborious searching for many tasks and saves a lot of time.

Finally, it is interesting that although the architectures searched with S1 and those transferred from meta-architecture searched with T-NAS are different for the specific tasks, their final evaluation performance is very close and is better than that of the random architectures. Such observation implies that some subspaces in architecture search space might be suitable for a specific task and T-NAS is able to adapt architecture initialized with $\widetilde{\theta}$ to the subspaces.

## 6 CONCLUSION AND FUTURE WORK

In this paper, we focus on the transferability of Neural Architecture Search, that is to say, how to get a suitable architecture for a new task in NAS? The two simple solutions are either inefficient or not optimal. To tackle this problem, we propose a novel Transferable Neural Architecture Search (T-NAS) for fast adaptation of architectures. Specifically, T-NAS learns a meta-architecture that is able to adapt to a new task easily and quickly through a few gradient steps, which is more flexible than the existing NAS methods. In addition, to optimize the whole search network, we propose an efficient first-order approximation algorithm. Extensive experiments show that T-NAS achieves state-of-the-art performance in few-shot learning setting. As for the supervised learning setting, T-NAS achieves comparable performance with other baselines but the searching cost is decreased by 50x, which demonstrates the effectiveness of our method.

For future work, we can study the transferability of NAS for those tasks from different task distributions, where some transfer learning methods might be helpful. We hope that this work can provide some insights on the transferability of NAS, which might potentially benefit the real-world applications.

**Acknowledgement.** The work is supported by the National Key R&D Program of China (2018AAA0100704) and the National Natural Science Foundation of China (NSFC) under Grants No. 61932020. We would like to thank Jiaxing Wang for some meaningful discussions.

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

## A  THE EXPERIMENTS OF AUTO-MAML

In Auto-MAML, we search a good network architecture for MAML. In fact, Auto-MAML is a special case of Method2 of Figure 1 in this paper, where all tasks share the same architecture searched with a meta-learning method. In the practical algorithm, it is similar to T-NAS, which is behaved as removing the update for $\theta$ in the meta-searcher stage. However, in Auto-MAML, we can divide the whole dataset into two splits for the updates of $\theta$ and $\widetilde{w}$ following the recent gradient-based NAS methods (Pham et al., 2018; Liu et al., 2018b). Here, the $\mathcal{D}_{\text{meta-train}}$ is divided into two independent splits $\mathcal{D}_{\text{train-split1}}$ and $\mathcal{D}_{\text{train-split2}}$ with $1 : 1$. The specific algorithm for meta-train and meta-test is shown in Alg. 3.

We follow the same definition for architecture search as T-NAS and we also use *one* {normal + reduction} cell for Auto-MAML. The searched architecture $\theta^*$ is shared by all tasks. We utilize the two splits of training data for architecture search. The search model is trained for 10 epochs with 5000 independent tasks for each epoch and the initial channel is set as 16. For base-searcher, we use the vanilla SGD to optimize the network weight $w_i^m$ with inner learning rate $\alpha_{\text{inner}} = 0.01$. The inner step $M$ is set as 5 for the trade-off between accuracy and time. For the meta-update, we use the Adam to optimize the network weights $w$ and architecture $\theta$ with outer learning rate $\alpha_{\text{outer}} = 10^{-3}$ and $\beta = 3 \times 10^{-4}$. The hyperparameter setting for network evaluation is the same as T-NAS. Here, we visualize some discrete architecture structure searched with Auto-MAML on the Mini-Imagenet dataset in Figure 3 and Figure 4.

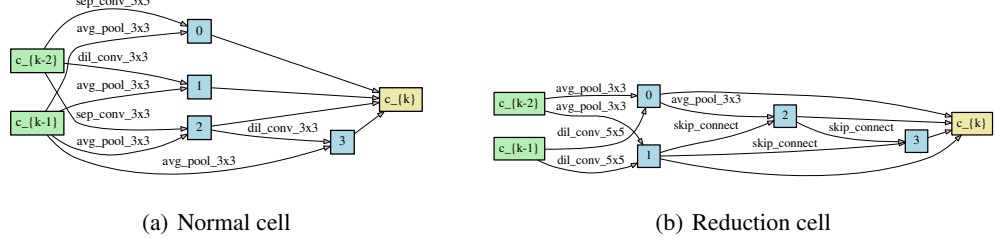

(a) Normal cell

(b) Reduction cell

Figure 3: Architecture searched with Auto-MAML in 5-way 1-shot setting of Mini-imagenet.

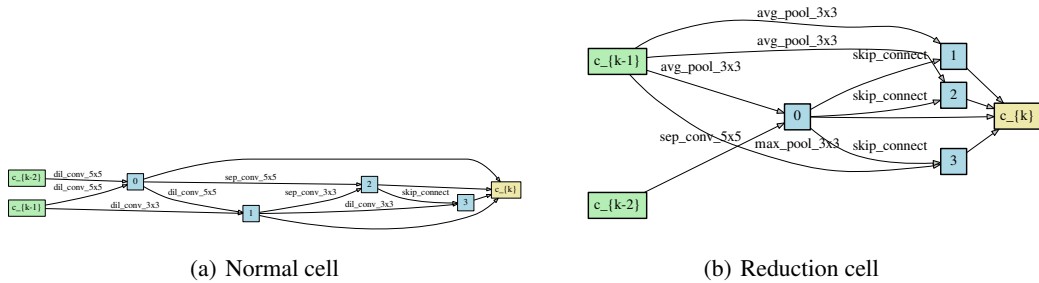

(a) Normal cell

(b) Reduction cell

Figure 4: Architecture searched with Auto-MAML in 5-way 5-shot setting of Mini-imagenet.

## B  TASK-SPECIFIC ARCHITECTURES

The aim of this paper is to learn a transferable architecture that is able to adapt to a new task through a few gradient steps. Therefore, it is meaningless that directly decoding for the searched meta-architecture $\widetilde{\theta}$ without regard to the specific tasks. Here, we visualize the (encoding of) transferable architecture $\widetilde{\theta}$ [4] searched with T-NAS and task-specific architecture $\theta^1, \theta^2, \theta^3$ in Figure 5. The matrix $(\widetilde{\theta}_{\text{normal}}, \widetilde{\theta}_{\text{reduce}})$ searched with T-NAS is smoother than the task-specific architecture matrices $(\theta^i_{normal}, \theta^i_{reduce})$, which shows that the meta-architecture is flexible and easy to adapt to these specific task ($\widetilde{\theta} \rightarrow \theta^1, \theta^2, \theta^3$).

## C  COMPLETE EXPERIMENTAL COMPARISON

In this section, we show the complete experimental comparison of our method with those methods using pretrained model in Table 6. Some methods (Oreshkin et al., 2018; Sun et al., 2019) get better performance by employing more complex networks and pretrained model.

## D  PERFORMANCE COMPARISON ON CIFAR-10 AND IMAGENET

To evaluate the transferability of our method, we also conduct the experiments on CIFAR-10 and ImageNet. Firstly, we construct a larger dataset from ImageNet to learn the meta-architecture, and then adapt the meta-architecture on CIFAR-10 to decode the final architecture. We test the performance of the final architecture on CIFAR-10 and ImageNet and report the performance in Table 7 and Table 8. From these two tables, we can see that the learned meta-architecture from T-NAS can quickly adapt to new tasks and achieve favorable performance. For example, given the learned meta-architecture from T-NAS, it only takes 0.042 GPU days to derive an architecture that achieves test error of 2.98% on CIFAR-10 and 27.2% on ImageNet. In contrast, searching for an architecture that achieves similar performance from scratch on CIFAR-10 by DARTS (first order) would cost

---

[4]It is represented with matrix as Liu et al. (2018b).

---

**Algorithm 3:** Auto-MAML

---

**Input:** Dataset $\mathcal{D}_{\text{train-split1}}$, $\mathcal{D}_{\text{train-split2}}$, inner learning rate $\alpha_{\text{inner}}$, outer learning rate $\alpha_{\text{outer}}$ and architecture learning rate $\beta$.

**Output:** The searched architecture $\theta^*$.

1 **% Meta-train:**
2 **while** *not done* **do**
3    **% Update** $w$
4    Sample batch of tasks $\{\mathcal{T}\}$ in $\mathcal{D}_{\text{train-split1}}$;
5    **for** $\mathcal{T}_i \in \{\mathcal{T}\}$ **do**
6       Get datapoints $\mathcal{T}_i^s$;
7       Compute $\nabla_{w_i^m}\mathcal{L}(g(\mathcal{T}_i^s; \theta, w_i^m))$ according to the standard cross-entropy loss;
8       Update $w_i^m$ with $w_i^{m+1} = w_i^m - \alpha_{\text{inner}}\nabla_{w_i^m}\mathcal{L}(g(\mathcal{T}_i^s; \theta, w_i^m))$ for $M$ steps;
9       Get datapoints $\mathcal{T}_i^q$ for meta-update;
10    **end**
11    Update $\widetilde{w}$ with $\widetilde{w} = \widetilde{w} - \alpha_{\text{outer}}\nabla_{\widetilde{w}}\sum_{\mathcal{T}_i \sim p(\mathcal{T})}\mathcal{L}_{\mathcal{T}_i}(g(\mathcal{T}_q; \theta, w_i^M))$ ;
12    **% Update** $\theta$
13    Sample batch of tasks $\{\mathcal{T}\}$ in $\mathcal{D}_{\text{train-split2}}$;
14    **for** $\mathcal{T}_i \in \{\mathcal{T}\}$ **do**
15       Get datapoints $\mathcal{T}_i^s$;
16       Compute $\nabla_{w_i^m}\mathcal{L}(g(\mathcal{T}_i^s; \theta, w_i^m))$ according to the standard cross-entropy loss;
17       Update $w_i^m$ with $w_i^{m+1} = w_i^m - \alpha_{\text{inner}}\nabla_{w_i^m}\mathcal{L}(g(\mathcal{T}_i^s; \theta, w_i^m))$ for $M$ steps;
18       Get datapoints $\mathcal{T}_i^q$ for meta-update;
19    **end**
20    Update $\theta$ with $\theta = \theta - \beta\nabla_\theta\sum_{\mathcal{T}_i \sim p(\mathcal{T})}\mathcal{L}(g(\mathcal{T}_i^q; \theta, w_i^M))$;
21 **end**
22 **% Meta-test:**
23 Sample tasks $\{\mathcal{T}\}$ in $\mathcal{D}_{\text{train-split2}}$;
24 **for** $\mathcal{T}_i \in \{\mathcal{T}\}$ **do**
25    Update $w_i^m$ with $w_i^{m+1} = w_i^m - \alpha_{\text{inner}}\nabla_{w_i^m}\mathcal{L}(g(\mathcal{T}_i^s; \theta, w_i^m))$ for $M$ steps;
26    Compute test accuracy $Acc_i$ in $\mathcal{T}_i^q$;
27 **end**
28 **Return** architecture $\theta^*$ according to the best average accuracy of $\{Acc\}$.

---

Table 6: 5-way accuracy results on Mini-Imagenet.

| Methods | Architectures | Parameters | 1-shot | 5-shot | Pretrained |
|---|---|---|---|---|---|
| TADAM (Oreshkin et al., 2018) | ResNet12 | 2039.2K | $58.5 \pm 0.3\%$ | $76.7 \pm 0.3\%$ | Y |
| MTL (Sun et al., 2019) | ResNet12 | 2039.2K | $61.2 \pm 1.8\%$ | $75.5 \pm 0.8\%$ | Y |
| Matching nets (Vinyals et al., 2016) | 4CONV | 32.9K | $43.44 \pm 0.77\%$ | $55.31 \pm 0.73\%$ | N |
| ProtoNets (Snell et al., 2017) | 4CONV | 32.9K | $49.42 \pm 0.78\%$ | $68.20 \pm 0.66\%$ | N |
| Meta-LSTM (Ravi & Larochelle, 2017) | 4CONV | 32.9K | $43.56 \pm 0.84\%$ | $60.60 \pm 0.71\%$ | N |
| Bilevel (Franceschi et al., 2018) | 4CONV | 32.9K | $50.54 \pm 0.85\%$ | $64.53 \pm 0.68\%$ | N |
| CompareNets (Sung et al., 2018) | 4CONV | 32.9K | $50.44 \pm 0.82\%$ | $65.32 \pm 0.70\%$ | N |
| LLAMA (Grant et al., 2018) | 4CONV | 32.9K | $49.40 \pm 1.83\%$ | - | N |
| MAML (Finn et al., 2017) | 4CONV | 32.9K | $48.70 \pm 1.84\%$ | $63.11 \pm 0.92\%$ | N |
| MAML (first-order) (Finn et al., 2017) | 4CONV | 32.9K | $48.07 \pm 1.75\%$ | $63.15 \pm 0.91\%$ | N |
| MAML++ (Antoniou et al., 2019) | 4CONV | 32.9K | $52.15 \pm 0.26\%$ | $68.32 \pm 0.44\%$ | N |
| Auto-Meta (small) (Kim et al., 2018) | Cell | 28/28 K | $49.58 \pm 0.20\%$ | $65.09 \pm 0.24\%$ | N |
| Auto-Meta (large) (Kim et al., 2018) | Cell | 98.7/94.0 K | $51.16 \pm 0.17\%$ | $69.18 \pm 0.14\%$ | N |
| BASE (Softmax) (Shaw et al., 2018) | Cell | 1200K | - | $65.40 \pm 0.74\%$ | N |
| BASE (Gumbel-Softmax) (Shaw et al., 2018) | Cell | 1200K | - | $66.20 \pm 0.70\%$ | N |
| Auto-MAML | Cell | 23.2/26.1 K | $51.23 \pm 1.76\%$ | $64.10 \pm 1.12\%$ | N |
| T-NAS | Cell | 24.3/26.5 K$^\star$ | $52.84 \pm 1.41\%$ | $67.88 \pm 0.92\%$ | N |
| **T-NAS++** | Cell | 24.3/26.5 K$^\star$ | $\mathbf{54.11 \pm 1.35\%}$ | $\mathbf{69.59 \pm 0.85\%}$ | N |

$^\star$ means the average parameters of architectures for evaluation.

1.5 days, which is about 36 times longer than that of T-NAS. This result confirms the advantage of T-NAS and also indicates that it is possible to apply T-NAS to practical scenarios.

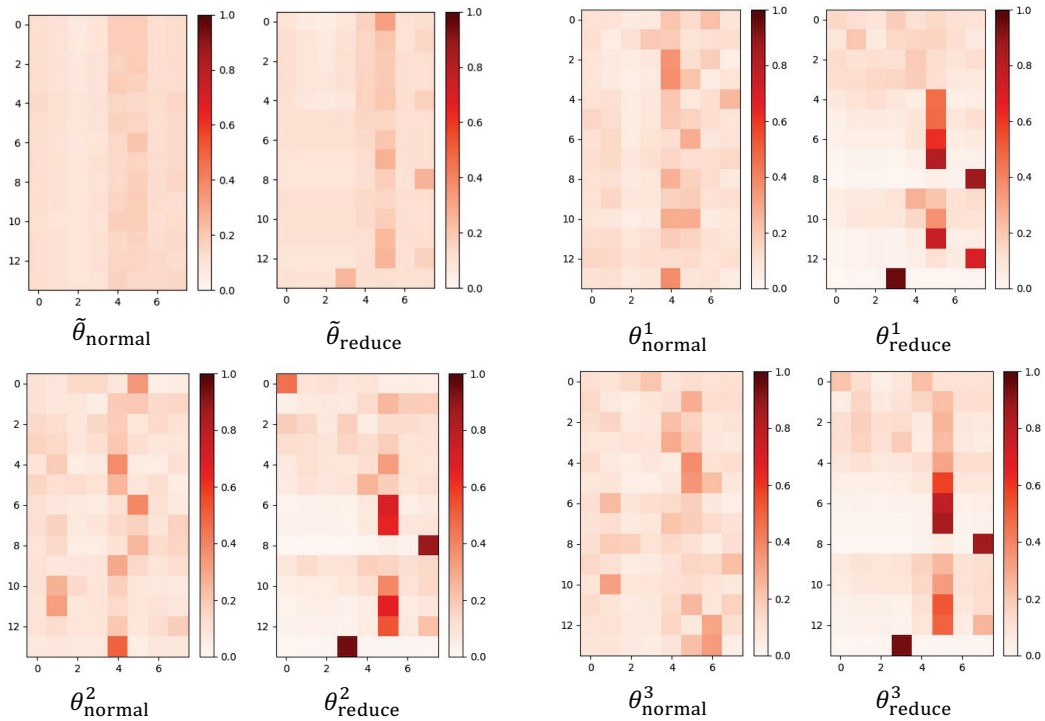

Figure 5: Meta-architecture matrix $(\widetilde{\theta}_{normal}, \widetilde{\theta}_{reduce})$ searched with T-NAS and three task-specific architecture matrices $(\theta^i_{normal}, \theta^i_{reduce})$. The search experiments are conducted in 5-way, 5-shot setting of Mini-Imagenet dataset.

Table 7: Comparisons with state-of-the-art image classifiers on CIFAR-10.

| Methods | Test Error (%) | #Param. (M) | Search Cost (GPU days) |
|---|---|---|---|
| Random search baseline + cutout | $3.29 \pm 0.15$ | 3.2 | – |
| NASNet-A + cutout (Zoph et al., 2018) | 2.65 | 3.3 | 180 |
| AmoebaNet-A + cutout (Real et al., 2018) | 3.34 | 3.2 | 3150 |
| AmoebaNet-B + cutout (Real et al., 2018) | $2.55 \pm 0.05$ | 2.8 | 3150 |
| PNAS (Liu et al., 2018a) | $3.41 \pm 0.09$ | 3.2 | 225 |
| ENAS+cutout (Pham et al., 2018) | 2.89 | 4.6 | 0.5 |
| DARTS (first-order) + cutout (Liu et al., 2018b) | $3.00 \pm 0.14$ | 3.3 | 1.5 |
| DARTS (second-order) + cutout (Liu et al., 2018b) | $2.76 \pm 0.09$ | 3.37 | 4 |
| Ours (first-order) + cutout | $2.98 \pm 0.12$ | 3.4 | 0.043 |

Table 8: Comparisons with state-of-the-art image classifiers on ImageNet in the mobile setting.

| Methods | Test Error (%) | | #Params (M) | Search Cost (GPU days) |
|---|---|---|---|---|
| | top-1 | top-5 | | |
| NASNet-A (Zoph et al., 2018) | 26.0 | 8.4 | 5.3 | 1800 |
| NASNet-B (Zoph et al., 2018) | 27.2 | 8.7 | 5.3 | 1800 |
| NASNet-C (Zoph et al., 2018) | 27.5 | 9.0 | 4.9 | 1800 |
| AmoebaNet-A (Real et al., 2018) | 25.5 | 8.0 | 5.1 | 3150 |
| AmoebaNet-B (Real et al., 2018) | 27.2 | 8.7 | 5.3 | 3150 |
| AmoebaNet-C (Real et al., 2018) | 27.5 | 9.0 | 4.9 | 3150 |
| PNAS (Liu et al., 2018a) | 25.8 | 8.1 | 5.1 | ~255 |
| DARTS (Liu et al., 2018b) | 26.9 | 9.0 | 4.9 | 4 |
| Ours | 27.3 | 9.0 | 4.9 | 0.043 |

