# OpenReview forum: "Towards Fast Adaptation of Neural Architectures with Meta Learning"
_ICLR.cc/2020/Conference — Accept (Poster)_

### Official Review · AnonReviewer1 · 2019-10-20
**Official Blind Review #1**

**Rating:** 6

**Review:**

Summary:

Current neural architecture search (NAS) methods work in the mode of what is called S1 in this paper: given a dataset, search for a new architecture from scratch for that particular dataset. This paper proposes using MAML-style metalearning for learning meta-architectures across many meta-training tasks so that given a new test task (dataset) the meta-architecture is a few search steps away from a near-optimal one. Hence the name Transferable-NAS (T-NAS).

The main method pretty much follows what one would expect for a MAML-style approach. During meta-training a number of tasks (datasets) are used to learn a meta-architecture and corresponding meta-weights (Algorithm 1) and then during meta-testing time, given a new task (dataset) use any NAS search technique starting from the meta-architecture and corresponding meta-weights. (The authors use DARTS as the NAS technique which by itself is the same bilevel optimization as MAML, but for NAS. This sets up a 4-level optimization problem during meta-training time!).

Experiments on two settings are presented: Few-shot learning and the more traditional supervised learning in NAS literature.

Comments:

- I really like the premise of the paper. While there have been papers trying to leverage dataset-level features for NAS transfer via bayesian approaches, computing dataset level features is always a bit difficult. The nice part about the MAML is that the transfer is embedded into the representation itself. But I have a bunch of clarification questions which is quite possible is mostly due to my misunderstandings. So please bear with me:

1. Section 5.2.1: "On the Mini-imagenet
dataset, One{normal + reduction} cell is trained 10 epochs with 5000 independent tasks for each epoch and the initial channel is set as 16." What are the 5000 independent tasks? In 5.1 it is said that 64 training classes, 16 validation classes and 20 test classes are present. I took that to mean that tasks are different classes of images in the few-shot setting. Clearly that is not the case. What precisely is a task in the few-shot learning case?

2. Secton 5.3: "...we choose 10 tasks with 200-shot, 50-query, 10-way for each task...." Again, what precisely is a task in the supervised learning setting?

3. One curious question I had is how much does architecture search matter as opposed to just taking the meta-architecture found by Alg 1 and just pretraining it on large image datasets like ImageNet or bigger and then just finetuning on downstream tasks. Specifically if you look at Table 3 ResNet12 pretrained has really good performance (although its size is ~100 times that of the others in the table). Perhaps S2 or AutoMAML with pretraining is the best option?

**Experience Assessment:**

I have published one or two papers in this area.

**Review Assessment: Checking Correctness Of Derivations And Theory:**

I assessed the sensibility of the derivations and theory.

**Review Assessment: Checking Correctness Of Experiments:**

I carefully checked the experiments.

**Review Assessment: Thoroughness In Paper Reading:**

I read the paper at least twice and used my best judgement in assessing the paper.

---

> ### Author Response · Authors · 2019-11-15
> **Re: Official Blind Review #1**
>
> Thank you for your careful and insightful comments. In the response, we have tried our best to address the questions. And the paper has also been modified accordingly.
>
> Q1: ‘This sets up a 4-level optimization problem during meta-training time!’
> A1: The naïve solution of T-NAS can be understood as a 4-level optimization problem. However, in order to make such optimization problem efficient, we treat $\theta$ and w equally as the parameters of the network function g. Such optimization is more efficient. This is also one of the contributions of this paper. We have clarified this point in the Sec.4.3 in the revised manuscript. Thanks for the comment!
>
> Q2: ‘What precisely is a task in the few-shot learning case?’
> A2: A task in few-shot learning can be constructed as follows: For example, for the 5-shot, 5-way setting, we firstly randomly sample 5 classes from the 64 training classes, and then randomly sample 5 images for each class to get a task. Thus, there are $5 \times 5$ = 25 images in each task. We repeat this process 5000 times to get 5000 independent tasks. Here, the definition of tasks is also consistent with previous meta-learning methods (Finn et al. 2017, Antoniou et al. 2019). We also add the detailed explanations in Sec.5.2 in the revised manuscript accordingly.
>
> Q3: ‘what precisely is a task in the supervised learning setting?’
> A3: For the supervised learning setting, the definition of tasks is the same as few-shot learning. The only difference between both is that we expand the number of training data from 1/5-shot to 200-shot to make the model can be trained from scratch.
>
> Q4: ‘how much does architecture search matter as opposed to just taking the meta-architecture found by Alg 1 and just pretraining it on large image datasets like ImageNet or bigger and then just finetuning on downstream tasks.’
> A4: Thanks very much! That is really an insightful question. To answer this question, three more experiments are conducted as suggested, which are:
> 1)	Auto-MAML (pretrained): The architecture searched by Auto-MAML is pretrained by following previous work (Oreshkin et al. (2018)) and finetuned on the current tasks.
> 2)	Meta-architecture (pretrained): Take the meta-architecture (after decoding) found by T-NAS and pretrain it, and then finetune on the current tasks.
> 3)	Meta-architecture (without pretraining): We only take the meta-architecture (after decoding) found by T-NAS and directly train it on the current tasks without pretraining.
> We compare the above settings with T-NAS, which adapts the meta-architecture to new tasks to produce the final architectures suitable for the new tasks without pretraining. The results are as follows:
>
> |                     Methods	                                    |            5-shot        |
> |           Auto-MAML (pretrained)	            |      69.42 ± 1.01% |
> |Meta-architecture (without pretraining)   |      62.14 ± 1.38% |
> |       Meta-architecture (pretrained)	    |      66.43 ± 1.11% |
> |                       T-NAS                                        |      67.88 ± 0.92% |
>
> We can see that
> 1)	T-NAS outperforms Meta-architecture (pretrained). From this result we can see that the architecture search is helpful. Actually, as the meta-architecture is optimized to be able to adapt to new tasks quickly, which is very smooth (c.f. Figure 2), thus is not suitable to be used directly. Even with pretraining, a better architecture can still achieve better performance.
> 2)	Meta-architecture (pretrained) performs better than Meta-architecture (without pretraining), which indicates that pretraining is important.
> 3)	Auto-MAML (pretrained) achieves the best performance in this experiment, which indicates that when the architecture is suitable enough, pretraining is always a good choice to improve the performance.
> At last, we would like to emphasize that T-NAS does not use pretraining at all. Actually, we could combine pretraining with T-NAS, e.g., first adapt the meta-architecture to different tasks and pretrain the new derived architectures, and then finetune ones on the new tasks. However, that means we have to pretrain for each task, which is too expensive to work in practice. However, we can see that, even without pretraining, the performance of T-NAS is still very good, which indicates the importance of architecture search.

---

### Official Review · AnonReviewer2 · 2019-10-23
**Official Blind Review #2**

**Rating:** 6

**Review:**

In this paper the author propose a combination of the neural architecture search method DARTS and the meta-learning method MAML. DARTS relaxed the search space and jointly learns the structural parameters and the model parameters. T-NAS, the method proposed by the authors, applies MAML to the DARTS model and learns both the meta-parameters and meta-architecture. The method is evaluated for the few-shot learning and supervised classification.
The idea is an incremental extension of MAML and DARTS. However, the idea is creative and potentially important. The paper and method are well described. The experimental results indicate a clear improvement over MAML and MAML++. A search time improvement at the cost of a small drop in accuracy is showns in Table 5. The experiments conducted follow closely the setups from the original MAML paper. Some more focus to the standard benchmarks for NAS would have been great. A comparison to the state-of-the-art in NAS on CIFAR-10 or ImageNet is missing.

**Experience Assessment:**

I have published one or two papers in this area.

**Review Assessment: Checking Correctness Of Derivations And Theory:**

I assessed the sensibility of the derivations and theory.

**Review Assessment: Checking Correctness Of Experiments:**

I assessed the sensibility of the experiments.

**Review Assessment: Thoroughness In Paper Reading:**

I read the paper at least twice and used my best judgement in assessing the paper.

---

> ### Author Response · Authors · 2019-11-15
> **Re: Official Blind Review #2**
>
> Thank you for your valuable comments. As suggested, the comparisons on the standard benchmarks (CIFAR-10 and ImageNet) have been added in the revision. Please see the response for details.
>
> Q1: ‘Some more focus to the standard benchmarks for NAS would have been great. A comparison to the state-of-the-art in NAS on CIFAR-10 or ImageNet is missing’.
> A1: Since the suggested experiment is to test the performance of the adapted architecture to much larger datasets (CIFAR-10 and ImageNet), we should learn the meta-architecture on a larger dataset. To this end, we construct the larger dataset from ImageNet to learn the meta-architecture, and then adapt the meta-architecture on CIFAR-10 to decode the final architecture. We test the performance of the final architecture on CIFAR-10 and ImageNet and report the performance in Appendix-D. From Table 7 and Table 8 in the revised manuscript, we can see that, the learned meta-architecture from T-NAS can quickly adapt to new tasks and achieve favorable performance. For example, given the learned meta-architecture from T-NAS, it only takes 0.042 GPU days to derive an architecture which achieves test error of 2.98% on CIFAR-10 and 27.2% on ImageNet. In contrast, searching for an architecture that achieves similar performance from scratch on CIFAR-10 by DARTS (first order) would cost 1.5 days, which is about 36 times longer than that of T-NAS. This result confirms the advantage of T-NAS and also indicates that it is possible to apply T-NAS to practical scenarios.

---

### Official Review · AnonReviewer3 · 2019-10-23
**Official Blind Review #3**

**Rating:** 6

**Review:**

This paper proposes a method, which is named T(Transferable)-NAS, for neural network architecture search (NAS) by leveraging the gradient based meta learning approach. The state of the art NAS algorithms search an architecture for a single task and a new architecture is searched from scratch for each new tasks. T-NAS learns a meta-architecture in order to adapt to the new tasks through a few gradient steps. This works uses MAML (Finn et al 2017) and some of its variants for gradient based meta learning. But instead of learning the weights (w), they learn a task sensitive meta architecture (\theta). To decode the learned \theta to task-specific architectures, they follow the method proposed by Liu et al 2018b. In that case, different architectures is used for different tasks as opposed to the baseline algorithms.

This paper proposes an incremental approach which is a combination of the existing algorithms. The most important contribution of this paper is providing Equation (5) that is used to update w and \theta parameters together by only backpropagating once. In this way, you don’t need the high-order derivatives and so you need less memory and search time. They compare T-NAS with state-of-the-art few-shot learning methods. I like the extensive empirical work of this paper. The prediction accuracy is mostly comparable with the baselines even some of the baselines are needed to be trained on many-shot classification tasks or use more complex architectures.

I think it would be good to add the method of Liu et al 2018b as a background. Because it is used as a part of the proposed algorithm and it is important to know the method to understand how the learned parameter \theta is adapted to the task-specific architectures. Besides that it is only mentioned once in the experiments section that T-NAS++ is based on MAML++ (which is an improved version of MAML, that investigates how to train MAML to promote the performance). If T-NAS++ is superior to the other baselines, including T-NAS, the difference should be clarified and emphasized more.

Table-3 presents the experiment results in a confusing way. In the current version, it is not very clear which method performs the best. For example I prefer to see the best results in bold. In addition, maybe it could be better to present the results separately for methods that uses pretrained models and do not use pretrained models. The reason why some methods perform better than the proposed method could be mentioned in the text instead of mentioning it as a footnote. Finally, it would be better for the readers to see the time comparisons explicitly given that one of the most important contribution of this work is to increase the efficiency.

**Experience Assessment:**

I have read many papers in this area.

**Review Assessment: Checking Correctness Of Derivations And Theory:**

I assessed the sensibility of the derivations and theory.

**Review Assessment: Checking Correctness Of Experiments:**

I carefully checked the experiments.

**Review Assessment: Thoroughness In Paper Reading:**

I read the paper at least twice and used my best judgement in assessing the paper.

---

> ### Author Response · Authors · 2019-11-15
> **Re: Official Blind Review #3**
>
> Thank you for your insightful and valuable suggestions. We have tried our best to improve our paper according to your comments. Please see the responses as follows.
>
> Q1: ‘I think it would be good to add the method of Liu et al 2018b as a background’.
> A1: We have added DARTS (Liu et al 2018b) to the background, including the formulation and the optimization of architecture parameter $\theta$. The specific contents can be found in Sec.3.2 of the revision. Thanks.
>
> Q2: ‘It is only mentioned once in the experiments section that T-NAS++ is based on MAML++. If T-NAS++ is superior to the other baselines, including T-NAS, the difference should be clarified and emphasized more’.
> A2: Thank you for the good suggestion. We have added the description of T-NAS++ in the last paragraph of Sec.4.2.
>
> Q3: ‘In the current version, it is not very clear which method performs the best. For example, I prefer to see the best results in bold’.
> A3: Thanks for the suggestion. In the revision, only the best results are in bold. Please check Table 2, Table 3, Table 4, and Table 5 for details.
>
> Q4: ‘In addition, maybe it could be better to present the results separately for methods that uses pretrained models and do not use pretrained models’.
> A4: Thanks for the suggestion. Since all the experiments (Auto-MAML, T-NAS, T-NAS++) are conducted without pretraining, we have separated the baselines that use pretrained model from Table 3 and moved them to the Appendix to make the comparison clearer. Please check Table 6 in the Appendix of the revision. Moreover, the reason for better performance is also mentioned in the text in Appendix-C as suggested.
>
> Q5: ‘It would be better for the readers to see the time comparisons explicitly given that one of the most important contribution of this work is to increase the efficiency’.
> A5: Thanks for the comment. Yes, increasing the efficiency is one of our contributions indeed, as the meta-architecture learned by T-NAS is able to adapt to a new task quickly. Such advantage is mainly demonstrated in the supervised learning scenario as it avoids searching from scratch for a new dataset. Please refer to Table 5 for the time comparison, where T-NAS achieves the competitive performance but with 50x less time cost (5 min vs. 266 min). For the few-shot learning scenario, the number of samples for each task is too small to support searching an architecture from scratch. Thus, the advantage of T-NAS is that the meta-architecture could adapt to a new task rather than using a fixed architecture for all tasks. In our experiments, this adaption costs 1.5 seconds (for 1-shot setting) and 7.8 seconds (for 5-shot setting), which is almost negligible compared with the accuracy improvement. Thanks for the suggestion, and some discussions are added in Sec.5.2.2 in the revised manuscript.

---

### Decision · Program_Chairs · 2019-12-19

**Decision:**

Accept (Poster)

**Comment:**

This paper introduces T-NAS, a neural architecture search (NAS) method that can quickly adapt architectures to new datasets based on gradient-based meta-learning. It is a combination of the NAS method DARTS and the meta-learning method MAML.

All reviewers had some questions and minor criticisms that the authors replied to, and in the private discussion of reviewers and AC all reviewers were happy with the authors' answers. There was unanimous agreement that this is a solid poster.

Therefore, I recommend acceptance as a poster.